# The Mechanisms of the Development of Atherosclerosis in Prediabetes

**DOI:** 10.3390/ijms22084108

**Published:** 2021-04-15

**Authors:** Yin Liang, Mengxue Wang, Chen Wang, Yun Liu, Keiji Naruse, Ken Takahashi

**Affiliations:** Department of Cardiovascular Physiology, Graduate School of Medicine, Dentistry and Pharmaceutical Sciences, Okayama University, Okayama 700-8558, Japan; pqjj7wl7@s.okayama-u.ac.jp (Y.L.); pmws31d7@s.okayama-u.ac.jp (M.W.); pneq0ew8@s.okayama-u.ac.jp (C.W.); pkfh3lxp@s.okayama-u.ac.jp (Y.L.); knaruse@md.okayama-u.ac.jp (K.N.)

**Keywords:** prediabetes, atherosclerosis, metabolic syndrome, obesity, endothelial dysfunction, exosome, microRNA

## Abstract

Lifestyle changes, such as overeating and underexercising, can increase the risk of prediabetes. Diabetes is one of the leading causes of atherosclerosis, and recently it became clear that the pathophysiology of atherosclerosis progresses even before the onset of diabetic symptoms. In addition to changes in platelets and leukocytes in the hyperglycemic state and damage to vascular endothelial cells, extracellular vesicles and microRNAs were found to be involved in the progression of prediabetes atherosclerosis. This review discusses the cellular and molecular mechanisms of these processes, with an intention to enable a comprehensive understanding of the pathophysiology of prediabetes and atherosclerosis.

## 1. Introduction

Contrary to the common belief that atherosclerosis is a disease of the elderly, the disease can begin surprisingly early, even before the age of 14 [1]. The modern age of gourmet foods, overeating, and underexercising has caused many people to develop metabolic syndromes that eventually lead to prediabetes and diabetes. Genetic factors probably play a role in this process. The likelihood of developing diabetes increases in subjects with a family history of diabetes [2]. Diabetes is one of the critical factors in the development of atherosclerosis. This fact is even relevant to the pandemic of coronavirus disease 2019 (COVID-19), as recent reports showed that patients with diabetes mellitus (DM) and atherosclerosis are at a greater risk for severe COVID-19 complications [3]. Therefore, it is essential to understand the mechanism underlying the development of atherosclerosis as diabetes progresses.

Numerous studies indicated that prediabetes can cause cardiovascular disease (CVD) [4,5]. Moreover, the burden of coronary atherosclerosis in prediabetic patients is more significant than that in normal people. Notably, the burden of atherosclerosis appeared even before the clinical manifestations of DM as seen in [4]. For example, lipid-rich plaques that congest blood vessels were already found in coronary lesions in the prediabetic state [4,6]. In addition, inflammation and vasoconstriction, which promote atherosclerosis in the coronary arteries, were observed in the prediabetic state [7,8]. Therefore, it is also meaningful to discuss inflammation and the role of immune cells in the pathogenesis of prediabetic atherosclerosis.

Moreover, it has become clear that the extracellular vesicles released from cells and microRNAs (miRNA) that regulate gene expression play important roles in the arteriosclerosis progression in prediabetes. Here, we discuss these exciting new findings on the mechanism of atherosclerosis in prediabetes to explore new ways of preventing the disease.

## 2. The Effect of the Release of Inflammatory Factors Caused by Obesity on CVD

Atherosclerosis is a chronic inflammatory process within the arterial wall [9]. Atherosclerosis is promoted when cholesterol-rich atherosclerotic Apo-B lipoproteins, such as low-density lipoproteins (LDL), very-low-density lipoproteins, and intermediate-density lipoproteins, enter into the subendothelial space via the endothelial passage (transcytosis) and accumulate in plasma [9]. The transcytosed Apo-B lipoproteins infiltrate macrophages and T cells to interact with the arterial wall cells, activating the inflammatory response [9,10]. Inflammation caused by obesity may accelerate atherosclerosis [11]. Adipose tissue is crucial in CVD caused by obesity [12]. In this situation, both fat cells and activated macrophages from the adipose tissue can produce various cytokines [12] (see Figure 1). These cytokines include adipokines induced by inflammation, such as leptin, adiponectin, tumor necrosis factor-alpha (TNF-α), interleukin 1 (IL-1), and IL-6; coagulants, such as PAI-1; vasoactive substances, such as leptin, angiotensinogen, and endothelin; and molecules such as FFA, TNF-α, and resistin that may cause insulin resistance. Table 1 shows the blood levels of several inflammatory factors associated with the development of atherosclerosis in prediabetes and diabetes.

IL-1 signaling involves the type I IL-1 receptor (IL-1R/IL-1R1) that can heterodimerize with the IL-1R accessory protein (IL-1RAcP) [13,14]. IL-1 receptor antagonist (IL-1Ra) is an anti-inflammatory cytokine that can compete with the proinflammatory cytokine IL-1 to bind to IL-1R. The relative occupation rate of the IL-1R1–IL-1RAcP receptor complex for IL-1 agonist or IL-1Ra controls the activation of the inflammatory signal [15,16]. In the obese condition, these cytokines released into the adipose tissue’s circulatory system can stimulate the production of C-reactive protein in the liver [17]. The level of the thrombosis molecule PAI-1 also increases before the development of obesity, whereas the level of adiponectin, which is only produced by fat cells, decreases [18]. Leptin is a crucial regulator of food intake and is a critical vasoactive substance produced by fat cells [19]. Other molecules produced by fat cells, including prostaglandin, adiponectin, and resistin, can influence metabolic function and lead to cardiovascular end-organ damage [20].

Regarding the effect of obesity on the cardiovascular system, researchers have also focused on the role of perivascular adipose tissue (PVAT). PVAT is an active endocrine organ that anatomically wraps the blood vessels and plays an important role in the pathogenesis of CVD. PVAT can secrete various adipokines, cytokines, and growth factors that inhibit or stimulate CVD development [21,22] (Figure 1). Dysfunction of PVAT can cause inflammation and oxidative stress and can decrease the production of vasoprotective adipocyte-derived relaxation factors and paracrine factors, such as resistin and leptin, and that of cytokines, such as IL-6 and TNF-α, and chemokines, such as RANTES and MCP-1 (also known as CCL5 and CCL2, respectively) [21]. These adipocyte-derived factors can trigger and coordinate the infiltration of inflammatory cells, such as T cells, B cells, and NK cells. Protective factors such as adiponectin can reduce the generation of NADPH oxidase superoxide and promote the bioavailability of nitric oxide (NO) in the blood vessel wall. Inflammatory molecules (such as IFN-γ or IL-17) can cause fibrosis in the endothelium, vascular smooth muscle cells, and adventitia, leading to the induction of vascular oxidase and eNOS dysfunction in cells [23,24]. These conditions lead to the occurrence of vascular dysfunction caused by dysfunctional perivascular fat. These mechanisms play a critical role in several CVDs, including atherosclerosis, hypertension, diabetes, and obesity [21,25].

## 3. Prediabetes Affecting Monocyte and Macrophage Activities That Lead to Atherosclerosis

For the progression of atherosclerosis, it is critical for the recruited monocytes to engulf LDL accumulated under the surface layer of blood vessels [34]. Hyperlipidemia in DM is known to increase the number of circulating neutrophils and monocytes through myelopoiesis [35].

Interestingly, Flynn et al. reported that even in the absence of DM or insulin resistance, the change in glucose level activated the bone marrow, leading to mononucleosis and accelerating atherosclerosis [36]. Meanwhile, transient intermittent hyperglycemia (TIH) caused hematopoietic stem and progenitor cells (HSPC) to proliferate and differentiate in the bone marrow, leading to a rapid expansion of common myeloid and granulocyte-macrophage progenitors and a subsequent white blood cell proliferation. In particular, TIH caused the proliferation of Ly6C^hi^ mononuclear cells, which induce atherosclerosis more strongly than other monocytes. According to Graubardt et al., removing the circulating Ly6C^hi^ monocytes reduces neutrophils that produce reactive oxygen species (ROS) [37], which is one of the critical factors of both DM and atherosclerosis [38]. Other risk factors found in prediabetic obesity, such as inflammation, dyslipidemia, oxidative stress, and signal transduction from adipose tissue macrophages, may also promote the expansion and differentiation of bone marrow HSPCs and, subsequently, leukocytosis [39,40,41].

On the other hand, glucose uptake by neutrophils after TIH is vital for driving CVDs [42,43]. Glucose transporter 1 (GLUT-1) is involved in glucose uptake; myeloid-restricted GLUT-1 deletion reduces glucose uptake by neutrophils and prevents TIH-induced myelopoiesis and atherosclerosis [36,43]. Flynn et al. found that when the glucose level is too high, neutrophils quickly reach their maximum glycolysis rate, leading to ROS production and protein kinase C (PKC) activation, which triggers the release of S100 calcium-binding proteins A8 and A9 (S100A8/A9). The release of S100A8/A9 initiates bone marrow production. S100A8/A9 proteins account for 40% of the neutrophil cytoplasmic protein, which is released upon inflammation and stimulates leukocyte recruitment, and it induces cytokine secretion [44,45]. S100A8/A9 is also upstream of the receptor for advanced glycation end-products (RAGE) signaling. The RAGE pathway plays a crucial role in hyperglycemia-induced myelopoiesis and TIH-induced mononucleosis. Koulis et al. discovered that RAGE mediation plays a vital role after a brief increase in blood sugar in atherosclerosis [46].

Another calcium-binding protein, S100A12, alias extracellular newly identified RAGE binding protein, is found in high concentrations in the blood of prediabetes and diabetes [26] (Table 1). As it facilitates the expression of adhesion molecules in the endothelial cells and the resulting migration of monocytes and macrophages, it is regarded as a pro-atherosclerotic factor [47]. Furthermore, the blood level of endogenously secreted RAGE, which can contribute to the neutralization of circulating RAGE ligands, is decreased in prediabetes [26]. This further facilitates the development of atherosclerosis. Therefore, targeting the RAGE pathway may be essential in eliminating hyperglycemia-induced atherosclerosis.

## 4. Relationship between Diabetes and Endothelial Dysfunction

While monocytes and macrophages differ in their roles in the pathogenesis of atherosclerosis, a loss of endothelial function is another factor that causes atherosclerosis. In the earliest stages of obesity, the endothelial function begins to be impaired.

### 4.1. Disrupted Endothelial Network Signaling in Prediabetes

Endothelial cells had long been considered a uniform cell group. However, it recently became clear that they have mosaic-like populations that respond differently and form a network to transmit information to change vascular permeability and perform remodeling functions [48]. The endothelial function manages the vascular function through multicellular network Ca^2+^ dynamics and the cellular heterogeneity [49].

Endothelial-dependent vasodilation is caused by network-level interactions between endothelial cell populations, and the prediabetic obesity’s control of this network-level can be damaged. Obesity changes the network activity and decreases the cell populations that respond to acetylcholine, causing vasodilation, increasing sensory cell clusters, and lengthening the distance for information transmission between cell clusters. For example, Wilson et al. demonstrated that communication between endothelial cells is disrupted in rats with prediabetic obesity [50]. They found that obesity impaired the calcium response in the endothelial cell network, leading to insufficient endothelial vascular tone control.

Multiple molecular mechanisms may cause abnormal vasodilation in prediabetes. The increase of ROS inactivates endothelial nitric oxide (NO), thus decreasing vasodilation in obese animals [51,52]. The potassium channel activity [53,54], NO release [54], or the regulation of endothelial function induced by the adipose tissue [55,56] may also lead to the impairment of endothelium-dependent vasodilation in obesity.

In summary, the decline in vascular function seen in prediabetes is due to abnormal signaling in the disrupted network of vascular endothelial cells, which may contribute to the development of atherosclerosis. These findings provide a new approach to analyze the vascular dysfunction in prediabetic obesity.

### 4.2. Endothelial Damage Induced by Microparticles

As early as 2010, researchers have discovered that microparticles (or submicron membrane vesicles) that are shed from the cellular plasma membrane are involved in endothelial damages that lead to atherosclerosis [57,58]. Moreover, Jansen et al. analyzed the circulating endothelial microparticles (EMP) produced by human coronary artery endothelial cells (HCAEC) at high concentrations of glucose. These EMPs were defined as “wounded” circulating endothelial microparticles (iEMP) and produced by “healthy” untreated HCAEC. Jansen et al. found that iEMP increased the activity of the ROS-producing NADPH oxidase; therefore, iEMP contained higher levels of ROS than EMP. iEMP triggered the phosphorylation of activating p38, a protein involved in apoptosis, in a manner depended on ROS. High glucose conditions could activate NADPH oxidase in EMPs, thereby exacerbating endothelial inflammation and impairing endothelial function. The microparticle-related endothelial damage and the impaired signaling of the endothelial network provide a framework to understand how endothelial cell damage exacerbates atherosclerosis in prediabetes.

### 4.3. Endothelial Dysfunction and Inflammation Increase

In 2016, researchers used enzyme-linked immunosorbent assays to analyze the levels of intercellular adhesion molecule-1 (ICAM-1), TNF-α, P-selection, and IL-6 in the serum of prediabetic patients. These markers are regarded as indicators of endothelial function and inflammation. The study results showed that the above four markers in patients with prediabetes were significantly higher than those in healthy subjects. Therefore, this suggested that prediabetes induces endothelial dysfunction and inflammation by increasing the levels of soluble adhesion molecules and inflammatory cytokines [59].

In addition, another study also explored endothelial dysfunction in diabetic patients. Researchers used the I-arginine test to compare the endothelial function and to measure the levels of TNF-α between healthy subjects and patients with prediabetes after a high-fat diet. This demonstrated that after a high-fat intake, the triglyceride and TNF-α levels in patients with prediabetes increased more than those without diabetes, and the endothelial function of diabetic patients was also reduced significantly [60].

Sardu et al. conducted clinical studies on the effects of biguanide treatment on reducing coronary endothelial dysfunction. The results showed that the percentage of endothelial dysfunction of the left anterior descending coronary artery (LAD) in patients with normal blood glucose was lower than that in patients with prediabetes. Following biguanide (metformin) application, the endothelial LAD dysfunction in patients with prediabetes was clearly eased compared with that in patients who had not been treated with biguanide. Furthermore, the incidence of adverse cardiovascular events was much higher in patients with prediabetes without biguanide treatment. Therefore, this indicates that biguanide application could reduce coronary endothelial dysfunction and decrease the risk of adverse cardiovascular events in prediabetes patients [61].

Current studies have shown that adiponectin’s serum level is related to coronary artery stenosis caused by impaired endothelial function [62,63]. Inhibition of adiponectin may lead to diabetes [64], obesity [65], atherosclerosis [66], and metabolic syndrome [18]. Ferdinando Carlo Sasso et al. conducted an observational study on the relationship between adiponectin, insulin resistance, and ischemic heart disease (IHD) in subjects with normal glucose tolerance [67]. This study performed an oral glucose tolerance test with subjects who have a history of diabetes or with impaired fasting blood glucose levels [67]. The scientists evaluated the relationship between adiponectin and insulin resistance and the occurrence of restenosis in these subjects, and findings showed that the level of adiponectin was directly related to the occurrence of restenosis [67]. In contrast, insulin resistance and adiponectin were independently associated with ischemic heart disease [67]. This study found that insulin resistance and adiponectin affected IHD at any stage in subjects with normal glucose tolerance [67].

## 5. Prediabetes-Facilitated Extracellular Vesicles Release in Atherosclerosis

Extracellular vesicles, which are spherical objects formed by exfoliation of cell membranes, consist of submicron-sized microparticles and nanometer-sized exosomes [68,69]. They affect the behavior of distant target cells, and in recent years they have generated much research interest [70,71,72]. The exosomes are involved in the pathogenesis of DM, including, specifically, insulin resistance [73]. Moreover, exosomes have been found to cause CVDs in diabetic conditions [74,75].

### 5.1. Platelet-Derived Microparticles in Prediabetic Patients

In 1993, Nomura et al. reported that platelets were activated and the concentration of plasma platelet-derived microparticles increased in patients with diabetes. Interestingly, a correlation was observed between the fasting blood glucose levels and the presence of platelet-derived microparticles [76]. Moreover, researchers found an association between the number of platelet-derived microparticles and the levels of LDL and triglycerides [77]; it was found that high levels of LDL and triglycerides are present in the development of atherosclerosis.

In 2014, Zhang et al. investigated the circulating microparticle concentrations from platelets, white blood cells, and monocytes in obese subjects and type 2 DM patients [78], and they found that the plasma concentration of platelet microparticle increased in type 2 DM patients regardless of their weight. Overall, these findings imply that a high level of blood glucose activates platelets and facilitates the release of platelet-derived microparticles, leading to the development of atherosclerosis.

### 5.2. CD36 Expressed in Platelets and Macrophages

CD36, initially characterized as a scavenger receptor at the platelet membrane that takes up lipoproteins, is involved in the pathogenesis of atherosclerosis [79,80]. CD36 is also expressed in macrophages and induces their transformation into foam cells, forming atherosclerotic plaques through their scavenging of oxidized LDLs [81] (Figure 1). Interestingly, Handberg et al. found a soluble form of CD36 in plasma and that the level of the plasma CD36 was correlated with fasting plasma glucose levels in type 2 DM patients [81]. Furthermore, according to Lopez-Carmona et al., CD36 is overexpressed in monocytes in atherosclerotic patients [82]. Another study indicated that circulating exosomes express functional CD36 and that the amount of exosomal CD36 increases in the postprandial state. Garcia et al. suggested that the circulating exosomes deliver free fatty acid in the bloodstream to cardiac tissue through an uptake via CD36 [83].

In summary, CD36, associated with prediabetes and atherosclerosis, is expressed at the cellular membranes of platelets and macrophages, in a soluble form, or exosomes. While CD36’s involvement in the pathogenesis of arteriosclerosis must be addressed in future research, its exosomal form can act as a marker for prediabetes and atherosclerosis.

### 5.3. CD105 Expression in Atherosclerosis and Prediabetes

Endoglin, also known as CD105, is a transmembrane glycoprotein mainly expressed on endothelial cells; it functions as a co-receptor for the transforming growth factor beta (TGF-β) [84]. In 2000, Bethell et al. reported the increased level of CD105 in the serum from atherosclerotic patients [85]. Later, Chironi et al. found that the thickness of intima-media in the common carotid artery was correlated with the increased level of CD105-expressing circulating microparticles, even before atherosclerosis was detectable [86].

In searching for the source of the microparticles expressing CD105 during the development of atherosclerosis, multiple research groups have identified cells with an unstable, vulnerable atherosclerotic plaque expressing CD105 [87,88,89]. During the development of atherosclerotic plaques, it was found that vasculature invades into the plaques. CD105 expression is associated with angiogenesis or neovascularization, where endothelial cells become activated and proliferate [90] (Figure 1). According to Luque et al., CD105 expression is associated with angiogenesis during the development of atheroma, but not necessarily with the atherosclerotic plaque itself [91]. CD105 expression was also observed in smooth muscle cells and macrophages in aortic atherosclerotic regions [92]. Therefore, the increase in CD105-expressing circulating microparticles during the progression of arteriosclerosis is likely due to their release from cells, especially endothelial, in atherosclerotic plaques.

Recent research has discovered that physical exercise affects the type and amount of extracellular vesicles in circulation [93,94]. Interestingly, Eichner et al. reported that interval exercise lowered the level of circulating extracellular vesicles expressing CD105 in prediabetic patients [95]. It is tempting to hypothesize that fitness prevents the development of atherosclerosis by reducing the number of circulating extracellular vesicles expressing CD105. However, the hypothesized involvement of CD105 in the development of atherosclerosis in prediabetes must be further investigated.

## 6. The Involvement of miRNAs in the Pathogenesis of Prediabetes and Atherosclerosis

In the previous section, we discussed the identification of extracellular vesicles express surface markers, such as CD36 and CD105, in prediabetic patients. These vesicles are likely involved in the development of atherosclerosis. CD36 is actively involved in fatty acid transport in macrophages. In addition to containing functional surface proteins, extracellular vesicles transport miRNAs in the bloodstream [96]. While some miRNAs are transported in an encapsulated form in vesicles, most miRNAs maintain their stability by forming a complex with other proteins during the transport in the bloodstream. This section discusses the mechanisms of miRNAs’ possible involvement in the pathogenesis of prediabetes and atherosclerosis.

### 6.1. Macrophage miRNAs Reducing the Production of Myeloid Cells and Suppressing Inflammation in Atheroma

Specific kinds of miRNAs induce anti-inflammatory effects. For example, the anti-inflammatory effect of *miR-146b* has been reported in a sepsis disease model and pneumonia [97,98]. In addition, *miR-146b* was reported to reduce inflammation via TRAF-6, a signal transducer in the NF-κB pathway, in a hypercholesterolemic condition that leads to atherosclerosis [99]. The expression level of *miR-146b* has been associated with HbA1c, a marker for high blood sugar level maintained over weeks/months [100]. Consistent with this finding, miR-146a expression is higher in DM patients than in prediabetic patients [101]. On the other hand, miR-99a, an anti-inflammatory miRNA, inhibits the production of inflammatory mediators, such as TNF-α, IL-6, IL-1β, and MCP-1 [102]. Moreover, the IL4-driven anti-inflammatory effects of miR-378a on macrophages have been suggested [103].

Interestingly, Bouchareychas et al. reported that macrophage exosomes ameliorated atherosclerosis by regulating hematopoiesis in the bone marrow and inflammation in atheroma through miRNA [104]. Mainly, the exosomes generated from naive bone marrow-derived macrophages (BMDM) contained anti-inflammatory *miR-99a*, *miR-146b*, and *miR-378a* [104]. It was found that exposure of IL-4 to BMDMs further facilitated the production of exosomes containing the anti-inflammatory microRNAs. These exosomal microRNAs inhibited the inflammation in atheroma by aiming at NF-κB and TNF-α signaling. Furthermore, the infusion of exosomes from IL-4 activated BMDM into mice can reduce hematopoiesis in bone marrow, thereby reducing both the number of macrophages in aortic root lesions and the number of atherosclerotic necrotic lesions.

The discovery that atherosclerosis development can be suppressed by exosomes containing miRNAs is quite sensational. Additionally, the high expression level of *miR-146b* in DM patients suggests that an anti-inflammatory response had already occurred in these patients. Thus, the administration of exosomes containing *miR-99a* and *miR-378a* to suppress the progression of atherosclerosis or inflammation is an anticipated clinical strategy.

### 6.2. Prediabetes Altering Expression of miRNAs in the Heart

Éva Sághy et al. found that prediabetes alters the expression of specific miRNAs that subsequently modulate their corresponding target mRNAs’ expression in the left ventricles of rats [105]. This study also found that *miR-141* and *miR-200c* were upregulated, and that *miR-200a*, *miR-208b*, and *miR-293* were downregulated in prediabetic patients. The authors predicted their target mRNAs computationally from the sequences of these miRNAs. They then verified the down-regulation of three mRNAs, i.e., the juxtaposed with another zinc finger protein 1 *(JAZF1)*, *Rap2c* of the RAS oncogene family, and zinc fingers with KRAB and SCAN domain 1 (*ZKSCAN1*). The miRNAs were isolated from homogenized ventricular tissue in this study. Therefore, it is necessary to investigate the origin of the detected miRNAs. It would also be interesting to see whether exosomes containing these miRNAs are released into the bloodstream. While the miRNAs may be released from noncardiomyocytes, such as cardiac fibroblasts or vascular endothelial cells, some studies suggested that cardiomyocytes secrete the exosomes, and that the effect of the exosomes is derived from miRNAs [106].

As mentioned above, *JAZF1* is one of the genes downregulated by miRNA in prediabetic patients. *JAZF1* is a transcriptional cofactor involved in gluconeogenesis, lipid metabolism, and insulin resistance; it is related to *JAZF1* expression in type 2 DM [107,108]. According to Li et al., *JAZF1* prevents atherosclerosis by reducing the number of atherosclerotic plaques in the aortic sinus [107]. Therefore, the decreased expression of *JAZF1* in prediabetes may be associated with the development of atherosclerosis.

### 6.3. miR-483 Causing Lipotoxicity, Insulin Resistance, and Impaired Endothelial Integrity

What we eat in the early stages of development affects the subsequent development of diabetes and atherosclerosis. Even more remarkably, the expression of a specific miRNA provides this effect. *miR-483* can be partially mediated by the translational growth inhibition/differentiation factor 3. It can also regulate the adipocytes’ ability to differentiate and store lipids [109]. Partially mediated by translational growth inhibition/differentiation factor 3, a target of *miR-483*, *miR-483* regulates the adipocytes’ ability to differentiate and store lipids. In summary, early life nutrition induces an increase in *miR-483* expression, which in turn limits lipid storage in adipose tissue, causing lipotoxicity and insulin resistance and increasing the risk of metabolic diseases.

In addition, *miR-483*’s expression level is higher in M2-type macrophages and in the aortic wall of patients with type 2 DM [110]. Moreover, overexpressing *miRNA-483* causes endothelial apoptosis and impairs endothelial regeneration [110]. Furthermore, serum *miR-483* level is associated with DM and CVD [111]. It would be interesting to identify the tissue that causes the elevation of serum *miR-483* and to examine whether the events occurring in endothelial cells and adipose tissue are associated with serum *miR-483*. In summary, excessive nutrition during early development may cause the upregulation of *miR-483*, leading to impaired endothelial integrity and subsequent CVDs. These possible effects of miR-483 upregulation should be examined in future studies.

## 7. Association between Prediabetes and Risk of Mortality Induced by CVD

Tang et al. compared the CVD-associated mortality between patients with prediabetes or those with diabetes. They found that there was no significant correlation between the risk of mortality from prediabetes and that of newly diagnosed diabetes. Patients with long-term diabetes were found to be at a high risk of mortality caused by CVD. Accordingly, the report suggested that in the elderly, long-term diabetes would lead to an increase in short-term mortality. The study also showed that at a median follow-up of 5.6 years, elderly people with prediabetes were still at a lower risk of death [112].

Additionally, Cai et al. investigated the association between prediabetes and the mortality and risk of CVD through a meta-analysis according to the Epidemiological Observational Research (MOOSE) group [113]. In this study, the authors found that prediabetes could increase mortality in patients with CVD or atherosclerotic heart disease [114]. Moreover, this study also showed that the risk of coronary heart disease and stroke would be higher in people with impaired glucose tolerance; the fasting blood glucose concentration for deaths caused by impaired fasting blood glycemia was 6.1–6.9 mmol/L [114]. Hence, this finding identified that prediabetes leads to an increase in mortality caused by cardiovascular disease.

## 8. Intervention and Treatment of Prediabetes

Prediabetes is a risk factor of T2DM and is also related to cardiovascular disease, so early interventions and treatments are critical [115]. Currently, lifestyle interventions, such as diet and exercise, are the preferred options [116,117]. In addition, medical therapy, such as treatment with biguanides (metformin), α-glucosidase inhibitors (acarbose), pancreatic lipase inhibitors (orlistat), PPAR-γ agonists (rosiglitazone, pioglitazone), and metatinib (nateglinide), is also extensively used to cure prediabetes [117,118]. Bariatric surgery is also suitable for patients with prediabetes who are also obese [119]. In 2019, scholars in the UK proposed that prediabetes treatment should be personalized according to the predictable phenotype’s specificity, and that interventions corresponding to that phenotype should be used. They suggested that knowledge of the phenotype’s specific pathophysiology may increase the value of funding appropriate treatments for patients with prediabetes [117]. This proposal provides new feasibility for intervention treatments for prediabetes patients and may optimize previous treatment plans.

## 9. Conclusions

Increasing attention is focused on understanding the mechanism that causes CVDs in prediabetes. As discussed above, hyperglycemia was found to cause increased hematopoiesis and ROS-producing neutrophils, leading to atherosclerosis development during prediabetes. These mechanisms were discussed in detail by Naoto Katakami in his review [120]. We further discussed the involvement of extracellular vesicles and miRNAs in the development of atherosclerosis in prediabetes. Hyperglycemia facilitates the production of extracellular vesicles from numerous types of cells, including vascular endothelial cells and leukocytes. These extracellular vesicles, including exosomes, facilitate atherosclerosis via specific proteins, such as ROS-producing NADPH oxidase and LDL-scavenging CD36. The extracellular vesicles also carry specific kinds of miRNAs that facilitate atherosclerosis by hematopoiesis and promoting inflammation.

The abovementioned functional proteins, miRNAs, and extracellular vesicles that transport miRNAs are highly anticipated targets for preventing or treating the development of atherosclerosis in prediabetes. On the other hand, it gives us hope that improving habits, such as diet and exercise, can suppress prediabetes and atherosclerosis via specific cellular mechanisms. In this era of increasingly prevalent prediabetes, further research in this area will contribute to the health of many.

## Figures and Tables

**Figure 1 ijms-22-04108-f001:**
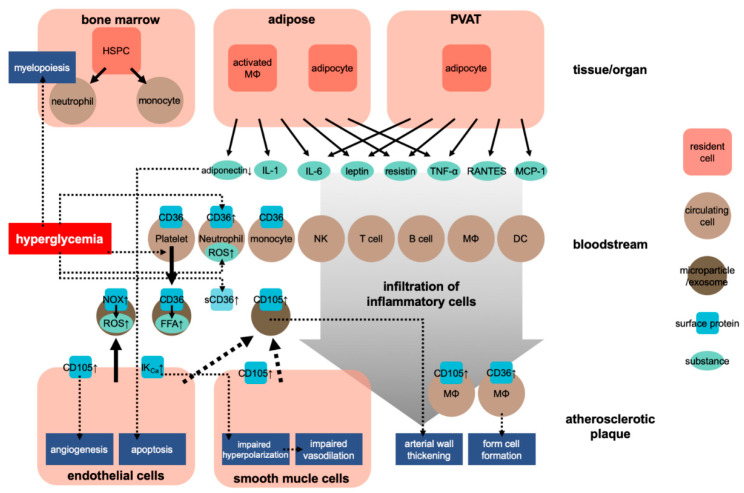
Development of atherosclerosis in obesity. HSPC, hematopoietic stem and progenitor cells; NK, natural killer cell; MΦ, macrophage; DC, dendritic cell; sCD36, soluble CD36; ↑, upregulation.

**Table 1 ijms-22-04108-t001:** Comparison of the factors contribute to atherogenesis between prediabetes and diabetes.

	Normal	Prediabetes	Diabetes	Refs
Fasting glucose (mg/dL)	86.4 ± 8.6	93.4 ± 11	120.2 ± 19	[26]
HbA_1c_ (%)	5.3 ± 0.2	6.0 ± 0.3	7.1 ± 0.5	[26]
Total cholesterol (mg/dL)	189–202	196–203	189–200	[26,27]
HDL cholesterol (mg/dL)	48–55	46–48	39–45	[26,27]
LDL cholesterol (mg/dL)	120–121	126–129	121–126	[26,27]
Triglycerides (mg/dL)	86–111	93–126	120–147	[26,27]
hs-CRP (mg/L)	1.4–2.1	2.4–3.4	4.0–4.5	[26,27]
esRAGE (ng/mL)	0.52 ± 0.26	0.32 ± 0.18	0.3 ± 0.19	[26]
S100A12 (ng/mL)	5.35 ± 3.38	7.13 ± 5.4	8.41 ± 4.44	[26]
Adiponectin (μg/mL)	9.52 ± 0.49	6.15 ± 0.49	6.57 ± 0.457	[28]
IL-6 (pg/mL)	1.77 ± 0.23	2.00 ± 0.14	2.84 ± 0.62	[29]
Resistin (ng/mL)	5.11 ± 1.56	9.16 ± 3.06	14.5 ± 5.31	[30]
TNF-α (pg/mL)	1.31 (0.69–2.25)	1.68 (0.79–2.01)	1.41 (0.83–1.86)	[31]
White blood cell (10^3^/μL)	6.4 ± 1.6	7.1 ± 1.8	7.2 ± 1.8	[26]
Intima-media thickness (mm)	0.67 (0.6–0.73)	0.75 (0.65–0.78)	0.78 (0.7–0.92)	[26]
Pulse wave velocity (m/sec)	7.1 ± 1.7	7.6 ± 1.6	8.6 ± 1.7	[26]
Coronary plaque progression (odds ratio)	-	1.338	1.635	[32]
Coronary artery calcification (odds ratio)	-	1.253	2.215	[33]
Atherosclerotic cardiovascular disease events (%)	14.24	17.81	30.40	[27]

HbA_1c_, glycated hemoglobin A_1c_; HDL, high-density lipoprotein; LDL, low-density lipoprotein; hs-CRP, high-sensitivity C-reactive protein; esRAGE, endogenous secretory advanced glycation end-products; IL-6, interleukin-6; TNF-α, tumor necrosis factor-α.

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
