# Peer review of "The Mechanisms of the Development of Atherosclerosis in Prediabetes"

_ijms, 2021, doi:10.3390/ijms22084108_

Round 1

Reviewer 1 Report

The mechanisms that favor the development of atherosclerosis in the prediabetic population are described in this narrative review. Prediabetes, like type 2 diabetes, is a condition whose prevalence is growing worldwide. The risk of CV complications in prediabetes is lower than in diabetes, but higher than in people with normoglycemia. Therefore, the interest in these pathophysiological mechanisms can have an important translational impact on clinical choices.

The paper is interesting and well written.

This reviewer raises only a few issues that the authors have to address.

In the Abstract, the statement "Lifestyle changes, such as overeating and under exercising, lead to prediabetes" is incorrect. Actually, lifestyle changes lead to overweight and obesity which are not always associated with diabetes. Therefore, this sentence should be corrected by adding "... in subjects with a predisposing genetic background". This point needs to be better clarified in the introduction too.

In the paragraph on the role of endothelial dysfunction in prediabetes, it would be appropriate to comment on the changes in inflammatory cytokines caused by meals (Nutr Metab Cardiovasc Dis. 2007 May;17(4):274-9. doi: 10.1016/j.numecd.2005.11.014.), as well as the protective effect of metformin on atheromatous damage in this patient setting (Diabetes Care. 2019 Oct;42(10):1946-1955. doi: 10.2337/dc18-2356).

Furthermore, the specific protective role played by adiponectin on atheromatous damage in non-diabetic insulin-resistant subjects deserves a comment (Cardiovasc Diabetol. 2019 Mar 4;18(1):24. doi: 10.1186/s12933-019-0826-0.).

The manuscript, like all pathophysiological reviews, requires 1-2 figures that can help the reader understand the mechanisms that link prediabetes and atherosclerosis. For example, a figure could schematize all the mechanisms involved, and another one in the detail of one of those described (such as endothelial dysfunction or extracellular vesicles or another chosen by the authors).

The bibliography is appropriate and up to date. However, there are some references from the 80s and early 90s that should be avoided in a review.

Author Response

We thank the reviewer for precious time for our manuscript. We hope our effort will meet the requirements of the high standards of the reviewer and science community.

In the Abstract, the statement "Lifestyle changes, such as overeating and under exercising, lead to prediabetes" is incorrect. Actually, lifestyle changes lead to overweight and obesity which are not always associated with diabetes. Therefore, this sentence should be corrected by adding "... in subjects with a predisposing genetic background". This point needs to be better clarified in the introduction too.

Thank you for the valuable suggestion. We have modified the abstract and introduction accordingly.

In the paragraph on the role of endothelial dysfunction in prediabetes, it would be appropriate to comment on the changes in inflammatory cytokines caused by meals (Nutr Metab Cardiovasc Dis. 2007 May;17(4):274-9. doi: 10.1016/j.numecd.2005.11.014.), as well as the protective effect of metformin on atheromatous damage in this patient setting (Diabetes Care. 2019 Oct;42(10):1946-1955. doi: 10.2337/dc18-2356).

Thank you for your suggestion. We have added a section concerning “4.3 Endothelial Dysfunction and Inflammation Increase.” This section discussed changes in inflammatory cytokines caused by food intake and the protective effect of metformin on atheromatous damage in this patient setting.

Furthermore, the specific protective role played by adiponectin on atheromatous damage in non-diabetic insulin-resistant subjects deserves a comment (Cardiovasc Diabetol. 2019 Mar 4;18(1):24. doi: 10.1186/s12933-019-0826-0.).

Thank you for your suggestion. This part has also been discussed in “4.3 Endothelial Dysfunction and Inflammation Increase.”

The manuscript, like all pathophysiological reviews, requires 1-2 figures that can help the reader understand the mechanisms that link prediabetes and atherosclerosis. For example, a figure could schematize all the mechanisms involved, and another one in the detail of one of those described (such as endothelial dysfunction or extracellular vesicles or another chosen by the authors).

Thank you for your suggestion. We added a figure that summarizes the mechanisms that link hyperglycemia and atherosclerosis and a table comparing the factors contribute to atherogenesis between prediabetes and diabetes.

The bibliography is appropriate and up to date. However, there are some references from the 80s and early 90s th.at should be avoided in a review.

Thanks for your suggestion. We have replaced these older references.

Reviewer 2 Report

Liang et al., covers the mechanisms of the development of atherosclerosis in pre- 2 diabetes.

Major comments:

  1. This is more or less having same type of information like the review by  Naoto Katakami J Atheroscler Thromb. 2018 Jan 1; 25(1): 27–39.
  2. The authors should cite the above review article, and discuss how this review adds more information than that review.
  3. As the review discusses about pre-diabetes, they should give one table showing the difference between Pre-diabetes and diabetes, as per as Atherosclerosis is concerned.
  4. The only novel information in this review is about  "involvement of miRNAs in the pathogenesis of prediabetes and atherosclerosis".
  5. There is no Figure or table given by the authors for the easy understanding.
  6. It would be good if the authors include some recent treatment plan and preventive measures for the same.

Author Response

We thank the reviewer for precious time for our manuscript. We hope our effort will meet the requirements of the high standards of the reviewer and science community.

This is more or less having same type of information like the review by  Naoto Katakami J Atheroscler Thromb. 2018 Jan 1; 25(1): 27–39. The authors should cite the above review article, and discuss how this review adds more information than that review.

Thank you for your suggestion. We cited the literature and mentioned what is new in our article in the conclusion.

As the review discusses about pre-diabetes, they should give one table showing the difference between Pre-diabetes and diabetes, as per as Atherosclerosis is concerned.

We added the table in accordance with the reviewer’s suggestion.

There is no Figure or table given by the authors for the easy understanding.

Thank you for your suggestion. We added a figure that discusses the mechanism of how pre-diabetes causes atherosclerosis.

It would be good if the authors include some recent treatment plan and preventive measures for the same.

Thank you for your suggestion. We have added this section regarding recent treatment plans for pre-diabetes.

Round 2

Reviewer 2 Report

Authors have responded adequately. No further query.